# The Anti-Inflammatory and Antioxidant Impact of Dietary Fatty Acids in Cardiovascular Protection in Older Adults May Be Related to Vitamin C Intake

**DOI:** 10.3390/antiox12020267

**Published:** 2023-01-25

**Authors:** Anna Gawron-Skarbek, Agnieszka Guligowska, Anna Prymont-Przymińska, Dariusz Nowak, Tomasz Kostka

**Affiliations:** 1Department of Geriatrics, Medical University of Lodz, 90-647 Lodz, Poland; 2Department of Clinical Physiology, Medical University of Lodz, 92-215 Lodz, Poland

**Keywords:** PUFA intake, Vitamin C intake, short-chain fatty acids, cardiovascular health, EPA, DHA, ALA, linoleic acid, inflammation, total antioxidant capacity

## Abstract

Polyunsaturated fatty acids, such as eicosapentaenoic acid (EPA), docosahexaenoic acid (DHA), α-linolenic acid (ALA), or linoleic acid (LA), have a particular role in counteracting cardiovascular diseases. They may regulate antioxidant potential and inflammatory reactions. Little is known whether other fatty acids, such as saturated fatty acids (e.g., short-chain fatty acids (SCFA) such as butyric or caproic acid) or monounsaturated fatty acids, may be involved and whether the level of Vitamin C intake may affect these processes. The purpose of this study was to assess the impact of fatty acid intake on plasma and salivary total antioxidant capacity (TAC), and the salivary inflammation marker C-reactive protein (CRP). Eighty older adults (60–79 years old) were divided into two groups with high (n = 39) and low (n = 41) Vitamin C intake. In the group with high Vitamin C intake SCFA, ALA, LA positively correlated with the plasma TAC indices, and in the group with low Vitamin C intake, the salivary TAC was decreased in subjects with a higher SCFA intake. Salivary CRP negatively corresponded to SCFA, EPA, and DHA in the whole study group (*p* < 0.05 for all). Fatty acids and Vitamin C intake may influence antioxidant potential and salivary CRP.

## 1. Introduction

Many studies have shown that the notion that fat is bad for the heart is wrong. The type of fatty acids is crucial, as well as the content of other nutrients in the diet. A diet based on good quality fats is generally beneficial for the proper functioning of the body, including the cardiovascular and nervous systems, or hormonal balance. There are two main families of polyunsaturated fatty acids (PUFA) that are relevant to human health, namely the ω-6 and the ω-3 PUFA. PUFA, such as eicosapentaenoic acid (EPA, 20:5 ω-3), docosahexaenoic acid (DHA), α-linolenic acid (ALA, 18:3 ω-3), linoleic acid (LA, 18:2 ω-6), or arachidonic acid (AA, 20:4 ω-6), have a particular role in counteracting cardiovascular diseases. They may regulate the antioxidant signaling pathway and modulate inflammatory processes [1,2,3]. EPA and DHA, in particular, have been found to produce both vaso- and cardio-protective responses via the modulation of membrane phospholipids, thereby improving cardiac mitochondrial function and energy production [4,5,6], lowering triglyceride concentrations, or inhibiting platelet activity [7].

Accumulating evidence has demonstrated that the administration of ω-3 PUFA or ascorbic acid following cardiac arrest improves survival. The antioxidant properties of Vitamin C can protect fatty acids; hence the assumption that a higher intake of fatty acids should be accompanied by a higher intake of Vitamin C (in practice, meals rich in fats, such as fish, dairy products, and vegetable oils, should be supplemented with raw fruit and vegetables, which are an important source of Vitamin C).

Nevertheless, little is known whether other fatty acids, such as saturated fatty acids (SFA) (e.g., short-chain fatty acids (SCFA) such as butyric or caproic acid) or monounsaturated fatty acids (MUFA), may be involved in these processes and whether the level of Vitamin C intake may affect them [8]. SCFA are able to restore imbalances in the lipid and glucose metabolism, and thus can make a contribution to the prevention and treatment of cardiovascular diseases [9]. They may be an effective fuel for the failing heart. Potential beneficial effects may also lie in the anti-inflammatory effects of SCFA, which has been shown to inhibit cardiac fibrosis—a key pathological process in the development of heart failure [10,11].

In terms of the cardioprotective effect of fats, we do not have complete knowledge whether, among the numerous known mechanisms, there is also one that is associated with the impact of fatty acids on the antioxidant barrier of the body, depending on the amount of Vitamin C, and thus, for example, enabling better functioning of the blood vessels or heart muscle. It is possible that the level of Vitamin C intake has no significant influence on antioxidant or anti-inflammatory processes, depending on the amount and type of fatty acid intake. Getting to know these relationships is especially important for the elderly, for whom, on the one hand, a higher intake of Vitamin C is recommended than for younger subjects and, on the other hand, it is often recommended to limit the amount of fat in the diet.

Therefore, the aim of the study was to assess the impact of the intake of fatty acids such as SCFA, MUFA, ω-3 and ω-6 PUFA on the plasma and salivary antioxidant capacity, and the CRP of saliva, depending on the level of Vitamin C intake, in the older adult population with a habitual diet.

## 2. Materials and Methods

### 2.1. Subjects

In total, 80 older (age range: 60–79 years) subjects at risk of cardiovascular disease (86% female) participated in the study. They were selected from a group of patients from the Department of Geriatrics of the Medical University of Lodz (Poland) who voluntarily participated in the healthy lifestyle workshops organized under the governmental program for the social activity of the elderly (2014–2020) and who underwent a detailed dietary and laboratory (blood plasma and saliva) assessment. The subjects were consecutively recruited on the basis of the inclusion criteria and their Vitamin C intake (see below) in order to obtain a balanced sex composition of the two groups that differed in Vitamin C intake.

None of the subjects was diagnosed with a tobacco addiction, active inflammatory processes (plasma CRP < 3 mg/L), renal dysfunction, disability, or dementia. None followed any special diet or supplementation. The study was approved by the local ethics committee (RNN/73/15/KE), and informed consent was obtained from each subject. The investigations were carried out following the rules of the Declaration of Helsinki of 1975, revised in 2008.

### 2.2. Study Design and Measurements

A detailed study protocol, including a description of laboratory tests and nutritional evaluation, has been described elsewhere [12]. The patients reported to the research center in the morning hours (between 8.00 and 10.00 a.m.), and fasting (at least 12 h) blood and unstimulated saliva samples were collected. A comprehensive assessment, including age, sex, anthropometric details, drug use, smoking, and dietary habits was performed for each subject [13]. A 24-h dietary recall from the day before the examination was obtained from each individual. The intake of energy, nutrients, vitamins, and minerals was calculated using the Diet 5.0 software package (developed by the National Food and Nutrition Institute, Warsaw, Poland).

The laboratory measurements included as part of the plasma study were the lipid profile parameters (total cholesterol, HDL-cholesterol, triglycerides), C-reactive protein (CRP), uric acid (UA), and four indicators of total antioxidant capacity (TAC), namely two in the native form (ferric reducing ability of plasma (FRAP) [14,15] and 2.2-diphenyl-1-picryl-hydrazyl (DPPH)) [15,16] and two in the non-urate form, i.e., after removing UA from the sample (non-urate-FRAP (Nu-FRAP) and non-urate-DPPH (Nu-DPPH)) [12].

In the saliva sample, salivary CRP, uric acid (SUA) and four TACs were determined, two in the native form (ferric reducing ability of saliva (FRAS) and 2.2-diphenyl-1-picryl-hydrazyl of saliva (DPPHS)), and two in the non-urate form: Nu-FRAS and Nu-DPPHS [12].

Both plasma and salivary TAC measurements were performed using spectrophotometric methods. The details of the methods are described elsewhere [12,15,17].

The recommendations of Polish scientific societies and the latest EFSA and FAO/WHO opinions were adopted to assess the level of compliance with the fat intake standards by the group, including the daily amount of energy from total fat, SFA, ALA, and LA, or the total intake of DHA and EPA, or cholesterol. The degree of insufficient intake of Vitamin C was estimated according to the standard RDA (recommended dietary allowance) for their age and sex (<75 mg and <90 mg per day for females and males over 60 years of age, respectively) [18].

Based on a median (Me) value of daily intake of Vitamin C, and taking the different interval of the values for sex into account, the study group was divided into Group 1 (n = 41) with a “high” Vitamin C intake (if the intake was ≥ Me) and Group 2 (n = 39) with a “low” Vitamin C intake (if the intake was < Me).

### 2.3. Statistical Analysis

To verify the normal distribution, a Shapiro–Wilk test was utilized. Variables that did not meet the assumption of normality were presented with the median value and quartile range (from Q_1_ to Q_3_) and analyzed by non-parametric statistics. The results for variables with a normal distribution were presented as the mean  ±  SD.

The quantitative characteristics between the groups with high and low Vitamin C intake were compared using the Mann–Whitney U-test because of the non-normally distributed data and by the chi^2^ test for comparing qualitative variables.

Spearman’s correlations were used to determine the relationships between fatty acid intake, the antioxidant indices in the plasma and the antioxidant parameters and CRP in the saliva. The level for statistical significance was set at *p* < 0.05. Statistica version 10 CSS software (StatSoft Polska Sp. z o.o., Kraków, Poland) was used for statistical analysis.

## 3. Results

### 3.1. Baseline Group Characteristics

The two subgroups did not differ with regard to age, sex distribution, anthropometrics and lipid profile characteristics (*p* > 0.05) (Table 1). Nearly 40% of the study group were diagnosed with obesity, the same percentage were diagnosed with overweight, and almost three-fourths (71%) presented with abdominal obesity. According to the body mass index (BMI) value, 18 subjects had first-degree obesity, 10 had second-degree obesity, and 3 had third-degree obesity. Overweight was recorded in 18 and 14 subjects from the low and high groups, respectively. None of the subjects was underweight. Based on the waist-to-hip ratios (WHR), taking the different norms for the sexes into account (WHR ≥ 0.8 for women and ≥ 1.0 for men), 64% of females and 8% of males were characterized by abdominal obesity.

The structures of both groups according to their nutritional status based on BMI, waist circumference, and WHR values did not statistically differ, though some tendency towards higher values in the low Vitamin C group should be noted.

### 3.2. Nutritional Characteristics

In the comparative analysis of fat intake, apart from higher cholesterol and total PUFA intake in the group with “high Vitamin C” intake, no other differences between the subgroups were found (Table 2).

The percentage of the study group with deficient Vitamin C consumption according to the RDA standard was 26% (21 subjects). No difference in Vitamin C intake was identified for sex (131 ± 89 mg for females vs. 136 ± 60 mg for males; *p* > 0.05). Almost three-quarters of the group (71%) covered the demand for 20–35% of energy derived from total fat, but in over the half of the group (in 44 subjects), the contribution of energy derived from SFA exceeded 10%. In terms of meeting the intake standards for selected PUFAs, in nearly half of the subjects (46% of the group), at least 0.5% of energy came from ALA, and the recommendations for LA intake (4% of energy from LA) and the combination of DHA and EPA (daily EPA+DHA intake of 250 mg) were met by 30% and 33% of the group, respectively. About 45% of the group consumed more than 300 mg of cholesterol in a day (data not shown in the table).

### 3.3. C-Reactive Protein of Saliva, and Plasma and Salivary Antioxidant Parameters

Table 3 presents the mean ± SD or the median (Q_1_-Q_3_) values of the native and non-urate plasma and salivary TAC indices, uric acid, and CRP. Nu-FRAP was lower in the group with a high Vitamin C intake; however, the other parameters did not differ between the groups.

### 3.4. Correlations for Fatty Acids Intake, and Plasma and Salivary Antioxidant Parameters

Several correlations were identified between antioxidant indices and fatty acids consumption, also in dependence on a level of Vitamin C intake.

#### 3.4.1. Antioxidant Parameters vs. Fatty Acid Intake in the Study Group (n = 80)

No relationships were found for any SFA or MUFA; however, the intake of some PUFA corresponded with a few antioxidant parameters.

The plasma antioxidant parameters usually did not correlate with fatty acids, except for positive correlations for DPPH (rho = 0.24) and Nu-DPPH (rho = 0.25) with ALA, and for DPPH (rho = 0.24) with LA (*p* < 0.05 for all). FRAP, Nu-FRAP, and plasma UA did not show a relationship with any PUFA intake.

Negative correlations were found for several salivary antioxidant indices: FRAS (rho = −0.24), Nu-FRAS (rho = −0.30), DPPHS (rho = −0.23), Nu-DPPHS (rho = −0.26), and AA only (*p* < 0.05 for all), but not for SUA.

#### 3.4.2. Plasma Antioxidant Parameters vs. Fatty Acid Intake in Groups with High and Low Vitamin C Intake

There were a few new positive correlations identified only in the group with a high Vitamin C intake for several SFA intake and some plasma antioxidant indices. Subjects with a higher butyric acid (4:0) intake were characterized by higher values of DPPH (rho = 0.40) and Nu-DPPH (rho = 0.41) (Figure 1a); however, the ones with a higher intake of caproic acid (6:0) had increased values of Nu-DPPH only (rho = 0.43) (*p* < 0.05 for all) (Figure 1b).

In the group with a low Vitamin C intake, the plasma antioxidant parameters did not correlate with MUFA or PUFA intake, except for single positive correlation for ALA intake and DPPH (rho = 0.35; *p* < 0.05), while in the group with a high Vitamin C intake, subjects with an increased intake of ALA had higher levels of FRAP (rho = 0.36) and DPPH (rho = 0.44) (Figure 1c), similar to those with an increased intake of LA (rho = 0.34 and rho = 0.44, respectively; *p* < 0.05 for both) (Figure 1d). Neither the non-urate plasma antioxidant indices nor UA showed a correlation with the level of consumption of any PUFA.

#### 3.4.3. Salivary Antioxidant Parameters vs. Fatty Acid Intake in the Groups with High and Low Vitamin C Intake

Some new negative correlations were revealed between the intake of some SFA and the salivary antioxidant indices in the group with a low Vitamin C intake as follows: between butyric acid and FRAS (rho = −0.37), DPPHS (rho = −0.41), and SUA (rho = −0.35) (Figure 2a), and between caproic acid and FRAS (rho = −0.35) and DPPHS (rho = −0.38) (*p* < 0.05 for all) (Figure 2b). These were not observed in the group with a high Vitamin C intake.

Of the few negative correlations with AA intake for the whole study group, in the group with a low Vitamin C intake, only the correlation with DPPHS (rho = −0.34) was observed, and in the group with a high Vitamin C intake, only the correlation with Nu-FRAS was noted (rho = −0.41) (*p* < 0.05 for both).

No other relationships were found between PUFA intake and salivary antioxidant indices in any subgroup.

### 3.5. Correlations of Fatty Acid Intake and Salivary CRP

Several negative correlations were identified between the inflammatory marker salivary CRP and the intake of some fatty acids in the whole study group: lower values for the CRP of saliva were observed in subjects with a higher intake of some SFAs, namely butyric (rho = −0.27) and caproic acid (rho = −0.26) (Figure 3a,b), and some PUFAs (especially DHA (rho = −0.28) and EPA (rho = −0.26)) (Figure 3c,d), as well as EPA + DHA (rho = −0.26) (*p* < 0.05 for all).

However, none of the above or any other new correlations for salivary CRP and fatty acid intake levels were found across the subgroups.

## 4. Discussion

Up to now, there has been no study that simultaneously considered antioxidant parameters (both plasma and salivary, and native and non-urate forms), the inflammatory marker CRP, and the intake of antioxidative Vitamin C and fatty acids. In the group with a high Vitamin C intake, SCFA, ALA, and LA positively correlated with the plasma TAC indices, and in the group with a low Vitamin C intake, salivary TAC decreased in subjects with a higher SCFA intake. Salivary CRP negatively correlated with SCFA, EPA, and DHA in the whole study group. Definitely positive correlations for plasma (non-urate and native) TAC markers with SCFAs (butyric or caproic acid) in the high Vitamin C group, and negative correlations for salivary (native) TAC markers with SCFA in the low Vitamin C group were noteworthy and indicated the possible contribution of Vitamin C to the observed relationships, as well as the significance of the study environment (comprehensive (plasma) and local (saliva)). It is all the more difficult to tell, with such a large set of variables, whether the contribution of UA to the obtained results (the correlations with native and non-urate TAC indices) also played a possible role. In this regard, the results of the analysis raise the question of whether plasma UA could mask the antioxidant effect of SCFA (correlations between Nu-DPPH and butyric or caproic acid). We can presume that along with a higher Vitamin C intake, SCFA, ALA or LA had a positive effect on plasma TAC. On the other hand, lower Vitamin C intake could cause losses of TAC in the saliva (lower FRAS) in subjects with a higher supply of SCFA. In order for food, which is an important source of SCFA, ALA, and LA, to have a positive effect on the antioxidant potential of the body, it should be consumed in the company of products rich in Vitamin C. For example, a butter sandwich should be eaten with raw vegetables, such as peppers, tomatoes, and onions, or a portion of fish should be eaten as part of dinner, accompanied by a salad. A higher consumption of SCFA, and EPA and DHA was associated with lower salivary CRP. Hence the assumption that the presence in the diet of products that are a source of SCFA (butter, dairy products) and ω-3 PUFA (fatty sea fish; in the diet of our seniors, the most popular were mackerel and herring) may have anti-inflammatory properties, regardless of the level of Vitamin C intake.

In trying to interpret the findings from the perspective of previous studies, we are limited to works that have focused, for example, on PUFA and their cardioprotective properties, but there are no studies that, similar to ours, show a complex system of dietary variables and their possible role in prophylaxis against cardiovascular diseases in terms of the mechanism based on the higher antioxidant and anti-inflammatory capacities of body fluids in humans. For instance, Li et al. examined the effect of DHA on cellular antioxidant capacity and mitochondrial function in HepG2 cells. Their findings suggested that DHA promoted mitochondrial function and biogenesis [19]. There are some molecular mechanisms underlying DHA’s function in improving resistance to and relieving the symptoms of chronic disease. A number of human studies have shown inverse associations between EPA and DHA status (e.g., the ω-3 index, which is the sum of EPA plus DHA in the erythrocytes) and blood markers of inflammation such as CRP or cytokines such as IL-6 [20,21,22]. Furthermore, meta-analyses of randomized controlled trials with EPA and DHA have confirmed reductions in the concentrations of CRP [23,24]. The ω-3 PUFA may lower the cardiovascular risk through a number of pleiotropic mechanisms, e.g., by lowering blood pressure, by mediating antithrombotic effects, by providing precursors for the synthesis of specialized pro-resolving mediators that can inhibit inflammation, or by modulating the lipid rafts enriched in cholesterol and sphingolipids [25]. Although ω-3 PUFA are stated to possess antioxidative properties, these molecules are highly oxidizable due to the multiple double bonds and may increase oxidative stress [26], counteracted by antioxidants, which are probably reduced as a result. According to the data of Cheng et al. (an animal study), the combination of ω-3 PUFA and Vitamin C treatment conferred an additive effect in suppressing lipid peroxidation and improving myocardial function [27]. On the other hand the anti-inflammatory, immunomodulatory, and antioxidative effects of ω-3 PUFA are in contrast to statements regarding ω-6 PUFA as promoting inflammation and suppressing cell-mediated immunity [28], which, in our study (LA), positively correlated with plasma TAC (FRAP). SCFA have been less discussed than long-chain fatty acids in cardiovascular diseases. However, increasing evidence indicates the importance of SCFA in regulating cardiac function [11]. SCFA are important in the cardiovascular system and their multiple effects in various pathophysiological processes have been noted, providing new insights into their promising clinical application, which was also discovered in the current study. A higher intake of SCFA (butyric and caproic acid) was associated with greater values of plasma TAC in subjects with a high Vitamin C intake (antioxidant effect), and with lower values of salivary CRP regardless of the level of Vitamin C consumption (anti-inflammatory effect). The data obtained in the study by Bartolomaeus et al. (animal model study) emphasized the immunomodulatory role of SCFA and their importance for cardiovascular health. Systemic inflammation was mitigated by propionate treatment, quantified as a reduction in splenic effector memory T cell frequencies and splenic T helper 17 cells, and a decrease in local cardiac immune cell infiltration in wild-type NMRI mice [29].

Despite its strengths, such as its complexity (simultaneously applying two analytical methods in two body fluids; using a number of parameters for the assessment; the age-, sex-, anthropometric- and nutrition-matched (fat intake) groups), our study also has some limitations. The relatively limited number of subjects and the cross-sectional design of the study preclude any categorical cause–effect statements.

Although the characteristics of both groups divided by nutritional status did not statistically differ on the basis of BMI, waist circumference, and WHR values, some tendency towards higher values in the low Vitamin C group should be acknowledged. Therefore, though the groups in the study were similar in terms of the frequency of the occurrence of people within the norm and outside the norm (according to BMI, waist circumference, and WHR), it would be worth conducting a similar study on a larger number of subjects, looking separately at the normal group only and the obese group, to eliminate the possible influence of a higher amount of adipose tissue on the relationships among Vitamin C intake, SCFA and PUFA intake, and the TAC and CRP levels. In other words, we need to assess whether normal subjects will show the same effects on the antioxidant potential, depending on greater amounts of fatty acids and Vitamin C being consumed, as ones with overweight and obesity.

It should be also noticed that our subjects were volunteers, who were probably healthier and fitter than a random sample, as well as more willing to participate in such studies. It was a relatively well-nourished group of older adults: most of them (nearly three-quarters of the group) followed the recommendations regarding the daily intake of Vitamin C or the amount of energy from fats, less than half followed the recommendations in terms of ALA, and about one-third followed them in terms of LA, EPA, and DHA intake.

## 5. Conclusions

Dietary fatty acids and Vitamin C may influence antioxidant potential and salivary CRP, so they may be an adjunct in cardiovascular protection. Food that is an important source of SCFA or essential PUFA may have a beneficial effect on the antioxidant potential of the body if consumed in the company of products rich in Vitamin C. However, the anti-inflammatory effect of SCFA and ω-3 PUFA may not depend on the level of Vitamin C intake. Further prospective and interventional human studies on a larger group of subjects are needed to examine this potential impact.

## Figures and Tables

**Figure 1 antioxidants-12-00267-f001:**
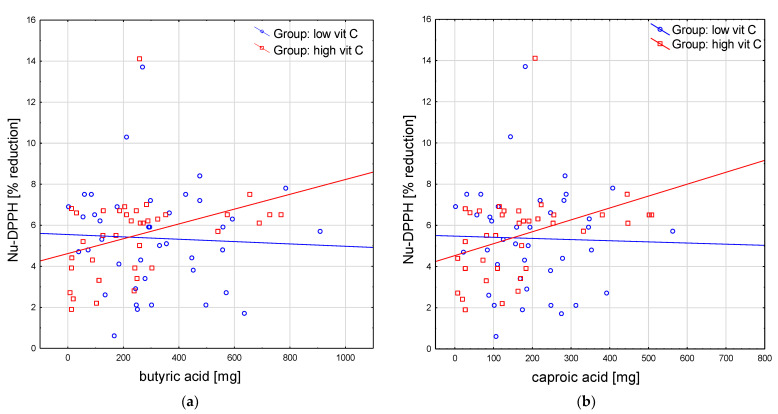
Selected correlations between plasma antioxidant parameters and short-chain fatty acids: (**a**) Nu-DPPH and butyric acid; (**b**) Nu-DPPH and caproic acid. Correlations between plasma antioxidant parameters with essential PUFAs: (**c**) FRAP and ALA; (**d**) FRAP and LA. ALA, alpha-linolenic acid; FRAP, ferric reducing ability of plasma; LA, linoleic acid; Nu-DPPH, non-urate 2.2-diphenyl-1-picryl-hydrazyl test of plasma.

**Figure 2 antioxidants-12-00267-f002:**
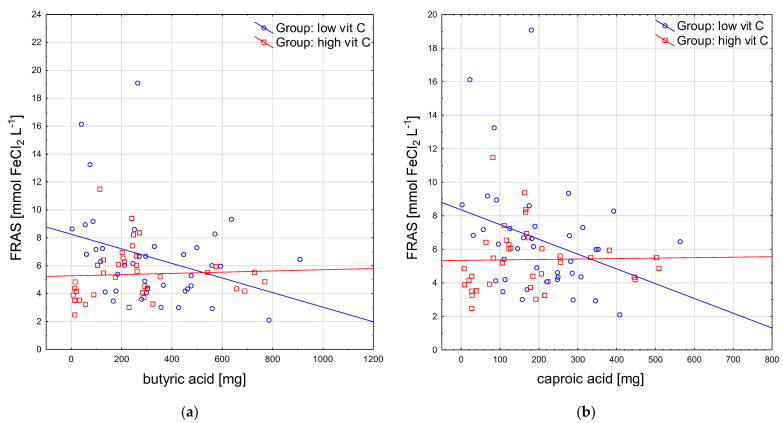
Selected correlations of salivary antioxidant parameters with short-chain fatty acids: (**a**) FRAS and butyric acid; (**b**) FRAS and caproic acid. FRAS, ferric reducing ability of saliva.

**Figure 3 antioxidants-12-00267-f003:**
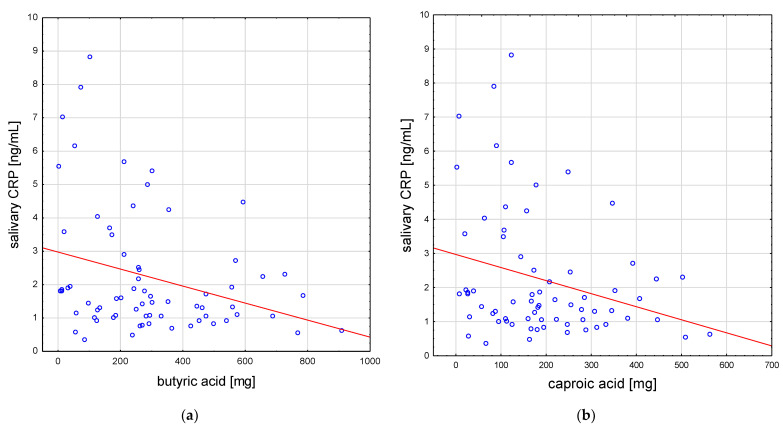
Selected correlations of salivary C-reactive protein with the short-chain fatty acids: (**a**) butyric acid and (**b**) caproic acid; and of salivary C-reactive protein with the PUFA: (**c**) DHA and (**d**) EPA in the whole study group. DPA, docosapentaenoic acid; EPA, eicosapentaenoic acid.

**Table 1 antioxidants-12-00267-t001:** Baseline characteristics of the study groups.

Variable		All (n = 80)	Low Vitamin C (n = 41)	High Vitamin C (n = 39)	*p*
Age (years)		66 (64–69)	66 (64–68)	67 (63–69)	NS
Sex (f/m)		69/11	35/6	34/5	NS
BMI (kg∙m^−2^)		28 (26–33)	29 (26–34)	27 (25–32)	NS
Waist circumference (cm)	f	89 (83–99)	92 (84–100)	88 (80–93)	NS
	m	99 (94–112)	105 (99–112)	94 (91–94)	NS
WHR	f	0.85 (0.80–0.90)	0.87 (0.80–0.91)	0.83 (0.78–0.88)	NS
	m	1.01 (0.96–1.04)	1.03 (1.01–1.05)	0.96 (0.86–0.96)	NS
TCh (mg dL^−1^)		187 (155–206)	181 (151–200)	190 (158–210)	NS
LDL–Ch (mg dL^−1^)		117 (86–135)	112 (84- 127)	127 (97–140)	NS
HDL–Ch (mg dL^−1^)		45 (34–54)	42 (33–52)	46 (43–57)	NS
TG (mg dL^−1^)		113 (77–170)	138 (93–174)	96 (73–149)	NS

Numerical data are presented as the median (Q_1_–Q_3_) and qualitative variables as the number of subjects. BMI, body mass index; WHR, waist-to-hip ratio; TCh, total cholesterol; LDL-Ch, low-density lipoprotein cholesterol; HDL-Ch, high-density lipoprotein cholesterol; TG, triglycerides; NS, non-significant difference; f, female; m, male.

**Table 2 antioxidants-12-00267-t002:** Comparison of the daily intake of selected nutrients between the study groups: Vitamin C, fatty acids, cholesterol, and energy from different fats.

Variable			All (n = 80)	Low Vitamin C (n = 41)	High Vitamin C (n = 39)	*p*
Vitamin C (mg)			114 (71–177)	77 (38–101)	177 (131–267)	<0.001
Total fat (g)			50 (36–64)	48 (37–57)	51 (36–66)	NS
Energy from fat (%)			29 ± 7	30 ± 8	28 ± 7	NS
Cholesterol (mg)			298 ± 166	258 ± 125	340 ± 193	0.02
SFA	Total (g)		18 (13–23)	17 (13–24)	18 (12–23)	NS
	Energy from SFA (%)		11 ± 3	11 ± 3	10 ± 3	0.04
	4:0 (mg)		255 (125–411)	292 (167–462)	240 (104–292)	NS
	6:0 (mg)		170 (92–255)	182 (107–282)	166 (73–215)	NS
	12:0 (g)		0.5 (0.2–0.6)	0.5 (0.3–0.7)	0.4 (0.2–0.6)	NS
	14:0 (g)		1.9 ± 1.0	2.0 ± 0.9	1.8 ± 1.0	NS
	16:0 (g)		9 (8–13)	9 (8–13)	10 (8–13)	NS
	18:0 (g)		3.8 (2.7–5.4)	3.9 (2.9–5.7)	3.5 (2.5–5.3)	NS
MUFA	Total (g)		19 (12–26)	19 (13–26)	20 (12–26)	NS
	Energy from MUFA (%)		11 (9–14)	11 (9–15)	11 (8–13)	NS
	16:1 (g)		1.1 (0.8–1.8)	1.1 (0.8–1.7)	1.1 (0.8–2.0)	NS
	18:1 (g)		17 (11–22)	17 (11–21)	17 (11–22)	NS
PUFA	Total (g)		7 (6–10)	6 (5–9)	7 (6–11)	0.04
	Energy from PUFA (%)		4 (3–5)	4 (3–5)	4 (3–6)	NS
	ω-3	ALA (g)	1.0 (0.6–1.4)	1.0 (0.8–1.4)	0.7 (0.6–1.5)	NS
		Energy from ALA (%)	0.5 (0.4–0.8)	0.4 (0.4–0.8)	0.5 (0.4–0.8)	NS
		EPA (mg)	9 (0–181)	5 (0–97)	17 (0–260)	NS
		DPA (mg)	6 (0–50)	10 (0–64)	4 (0–48)	NS
		DHA (mg)	63 (16–358)	35 (14–271)	70 (33–404)	NS
		EPA + DHA (mg)	70 (21–592)	47 (18–368)	81 (35–732)	NS
	ω-6	LA (g)	5.5 (4.0–7.5)	5.2 (3.9–6.8)	6.0 (4.7–9.0)	NS
		AA (mg)	104 (49–192)	104 (60–175)	110 (42–238)	NS

Data are presented as the mean ± SD for variables with a normal distribution and as the median (Q_1_–Q_3_) for non-parametric variables. NS, non-significant difference; SFA, saturated fatty acids (4:0, butyric acid; 6:0, caproic acid; 12:0, lauric acid; 14:0, myristic acid; 16:0, palmitic acid; 18:0, stearic acid); MUFA, monounsaturated fatty acids (16:1, palmitoleic acid; 18:1, oleic acid); PUFA, polyunsaturated fatty acids (ω-3: 18:3, ALA (alpha-linolenic acid); 20:5, EPA (eicosapentaenoic acid); 22:5, DPA (docosapentaenoic acid); 22:6, DHA (docosahexaenoic acid); ω-6: 18:2, LA (linoleic acid); 20:4, AA (arachidonic acid).

**Table 3 antioxidants-12-00267-t003:** Plasma and salivary non-urate and native antioxidant indices, and C-reactive protein for the groups with high and low Vitamin C intake.

Variable			All (n = 80)	Low Vitamin C (n = 41)	High Vitamin C (n = 39)	*p*
plasma	TAC,	FRAP (mmol FeCl_2_ L^−1^)	1.2 ± 0.2	1.3 ± 0.2	1.2 ± 0.2	NS
		DPPH (% reduction)	23 ± 6	24 ± 6	22 ± 5	NS
		Nu-FRAP (mmol FeCl_2_ L^−1^)	0.40 (0.36–0.45)	0.43 (0.37–0.52)	0.38 (0.36–0.43)	0.009
		Nu-DPPH (% reduction)	6 (4–7)	6 (4–7)	6 (4–7)	NS
	UA (mg dL^−1^),		4.5 ± 1.2	4.5 ± 1.0	4.4 ± 1.3	NS
	CRP (mg L^−1^)		<3	<3	<3	-
salivary	TAC,	FRAS (mmol FeCl_2_ L^−1^)	5.5 (4.2–6.8)	6.1 (4.3–7.3)	5.2 (4.1–6.3)	NS
		DPPHS (% reduction)	24 (16–35)	22 (16–32)	27 (16–37)	NS
		Nu-FRAS (mmol FeCl_2_ L^−1^)	1.29 (0.96–1.69)	1.37 (1.08-–1.66)	1.22 (0.96–1.74)	NS
		Nu-DPPHS (% reduction)	5 (4–7)	4 (3–7)	5 (4–8)	NS
	UA (mg dL^−1^)		8.7 (5.9–11.7)	8.7 (6.5–11.5)	8.5 (5.6–11.3)	NS
	CRP (ng mL^−1^)		1.6 (1.1–2.6)	1.3 (1.0–1.9)	1.8 (1.1–3.5)	NS

Data are presented as the mean ± SD for variables with a normal distribution and as the median (Q_1_-Q_3_) for non-parametric variables. CRP, C-reactive protein; Nu-FRAS, non-urate ferric reducing ability of saliva; FRAS, ferric reducing ability of saliva; Nu-FRAP, non-urate ferric reducing ability of plasma; FRAP, ferric reducing ability of plasma; Nu-DPPHS, non-urate 2.2-diphenyl-1-picryl-hydrazyl test of saliva; DPPHS, 2.2-diphenyl-1-picryl-hydrazyl test of saliva; Nu-DPPH, non-urate 2.2-diphenyl-1-picryl-hydrazyl test of plasma; DPPH, 2.2-diphenyl-1-picryl-hydrazyl test of plasma; TAC, total antioxidant capacity; UA, uric acid; NS, non-significant difference.

## Data Availability

Not applicable.

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
