# Peer review of "The Anti-Inflammatory and Antioxidant Impact of Dietary Fatty Acids in Cardiovascular Protection in Older Adults May Be Related to Vitamin C Intake"

_antioxidants, 2023, doi:10.3390/antiox12020267_

Round 1
Reviewer 1 Report
Dear Authors,
Thank you for the opportunity to review this interesting for me paper.
The aim of the study was to assess the impact of fatty acids (short-chain SFA, MUFA, ω-3 and ω-6 PUFA) intake on plasma and salivary antioxidant capacity, and CRP of saliva, in the dependence on the level of vitamin C intake, in the older adults population with a habitual diet. In my opinion, research on the potential cardioprotective effect of a habitual diet, especially in the elderly population, is highly desirable from the point of view of population health of an aging society. So the publication considers an interesting and topical research problem and the conclusions may have of significant application and public health importance. The manuscript of the publication is essentially correct. The theoretical chapters were well written. My comments mainly concern the description of the research material (subjects) and the presentation of the results.
Comments and Suggestions for Authors:
Section 1. Introduction.
1. The chapter is well written in my opinion.
Section 2. Materials and Methods.
2. The age range of the subjects given in the abstract of the publication and then in the text in Section 2.1 (line 68) clearly differs from the age of the subjects given in Table 1. Basic characteristics. Please clarify and unify these entries.
3. Please describe in detail in the upper row of the tables (applies to all tables) what values are presented in the tables. E.g. what is given in brackets, is it a minimum, maximum range? Please provide appropriate additions.
4. Thank you for the detailed description of the inclusion and exclusion criteria and the information on obtaining the approval of the ethics committee.
Section 3. Results.
5. I would like to point out that the only significant difference in the basic characteristics (Table 1) concerns the WHR index. This indicator is related to the unhealthy distribution of adipose tissue. The table shows that also WC (despite the surprising lack of statistical significance) clearly differs in the low vs. high vit C intake groups. The table shows that the high vit c intake group was clearly slimmer (BMI - 2 units less), lower WC (7 units less), lower WHR, lower TG concentration (42 units!). A clear somatic differentiation of the compared groups can significantly affect the results, can't it? Please provide detailed information in table 1 on the frequencies of the respondents in terms of BMI (deficiency, normal, overweight, obesity 1st, 2nd, 3rd) and the frequency of people in terms of WHR regularity (calculated separately for both sexes - other ranges of norms).
Section 4. Disscusion.
6. In my opinion, the Limitation study section should be expanded, especially in the context described in the point above.
Additional remarks, of minor importance:
7. please unify the fonts (e.g. line 163-166),
8. the citation should be in square brackets,
9. the names of the publications are not precisely given, e.g. 1, 2, 3, ..., the publication number is missing - please provide appropriate supplement.
Thank You.
Reviewer 2 Report
Dear Authors,
The reviewed paper entitled: "The Anti-Inflammatory and Antioxidant Impact of Dietary Fatty Acids in Cardiovascular Protection in Older Adults May Be Related to Vitamin C Intake" is valuable for this area of research, as the authors made important discoveries. Among other things, they found new negative relationships between SFA intake and antioxidant indices of saliva in the group with low vitamin C intake, i.e., between butyric acid and FRAS (DPPHS and SUA, and between caproic acid and FRAS and DPPHS. Therefore, this work deserves publication. No however, this means that it is excellent, but it does require a few changes and additions. Here are the details:
1. The abstract should be better structured and shortened.
2. In the introduction, the aim of the work should be clearly defined and an alternative research hypothesis to the null hypothesis should be put forward. The verification of the hypothesis should be carried out later in the work, preferably in the "Discussion" chapter.
3. The authors should separate the "Discussion" chapter and discuss the research results with other authors much better, using the latest literature.
4. Conclusions should be general and summarizing. They should be supplemented with at least one practical proposal - as nutritional recommendations and one proposal for the future
5. Supplement the list of literature with the latest items from 2021-2023.
